# Trimethylamine N-Oxide as a Biomarker for Left Ventricular Diastolic Dysfunction and Functional Remodeling After STEMI

**DOI:** 10.3390/ijms26073400

**Published:** 2025-04-05

**Authors:** Tsung-Ying Tsai, Ali Aldujeli, Ayman Haq, Paddy Murphy, Ramunas Unikas, Faisal Sharif, Scot Garg, Emmanouil S. Brilakis, Yoshinobu Onuma, Patrick W. Serruys

**Affiliations:** 1CORRIB Research Centre for Advanced Imaging and Core Lab, University of Galway, H91 TK33 Galway, Ireland; 2Cardiovascular Center, Taichung Veterans General Hospital, Taichung 407219, Taiwan; 3Institute of Cardiology, Lithuanian University of Health Sciences, 44307 Kaunas, Lithuaniaramunas.unikas@kaunoklinikos.lt (R.U.); 4Abbott Northwestern Hospital/Minneapolis Heart Institute Foundation, Minneapolis, MN 55407, USA; haq.ayman@gmail.com (A.H.); esbrilakis@gmail.com (E.S.B.); 5Department of Cardiology, University Hospital Galway, University of Galway, H91 YR71 Galway, Ireland; 6Department of Cardiology, Royal Blackburn Hospital, Blackburn BB2 3HH, UK; 7School of Medicine, University of Central Lancashire, Preston PR1 2HE, UK

**Keywords:** trimethylamine N-oxide (TMAO), gut microbiota, ST-elevation myocardial infarction, left ventricular functional remodeling, left ventricular diastolic dysfunction

## Abstract

Despite successful primary percutaneous coronary intervention (PPCI), the incidence of heart failure (HF) following ST-elevation myocardial infarction (STEMI) remains high. We investigated using Trimethylamine N-oxide (TMAO), a gut microbiota-derived biomarker, to predict adverse functional left ventricular (LV) remodeling (FLVR) and/or diastolic dysfunction (DD), which are precursors of HF post-STEMI. This prospective, observational study enrolled 204 STEMI patients with multivessel coronary artery disease after PPCI. TMAO level was collected at the baseline and 3 months. An echocardiography was performed at the baseline and at 12 months. The primary endpoints were the number of patients developing Group 4 FLVR or ≥Grade II DD at 12 months. The median age was 65 [57.00, 76.00] and 39.7% were women. The primary endpoints occurred in 47 (23.0%) patients. Three months of TMAO can discriminate patients with/without ≥Grade II LV DD and FLVR Grade 4 with areas under the curve (AUC) of the ROC of 0.72 (95% CI: 0.63–0.81; *p* < 0.001) and 0.77 (95% CI: 0.63–0.91), respectively. Similar results were shown in the validation cohort of 31 patients. The addition of 3 months of TMAO to traditional risk factors significantly improved the AUCs from 0.675 to 0.736 for ≥Grade II DD and from 0.793 to 0.873 for FLVR Grade 4. In multivariable logistic regression, 3 months of TMAO was independently associated with ≥Grade II DD (OR: 1.29 (1.13–1.50), *p* < 0.001) and FLVR Grade 4 (OR: 1.28 (1.12–1.47), *p* < 0.001). Three months of TMAO is strongly associated with LV DD and adverse remodeling after STEMI and may help identifying such patients for early treatment.

## 1. Introduction

Heart failure (HF) following ST-elevation myocardial infarction (STEMI) triples mortality rates and remains prevalent despite timely primary percutaneous coronary intervention (PPCI) [1]. While the incidence of pre-discharge HF has declined over the last decade, over 50% of post-STEMI HF diagnoses occur post-discharge, with nearly half of these cases being heart failure with a preserved ejection fraction [2]. Notwithstanding this, the prognosis of HF is similar regardless of the left ventricular ejection fraction (LVEF), whereas HF developing after discharge confers a worse prognosis than an in-hospital diagnosis [3].

Functional left ventricular remodeling (FLVR) and left ventricular diastolic dysfunction (LVDD) are precursors to clinical HF [4,5], representing the complex interaction between risk factors, inflammation, fibrosis, hemodynamics, and coronary microvascular dysfunction (CMD) [6,7]. Early and aggressive pharmacological treatment is key to blunting and even reverting the disease process [8]. Consequently, the early identification of high-risk patients is vital, preferably with simple and cost-effective biomarker measurement [7].

Trimethylamine N-oxide (TMAO) is a novel biomarker derived from gut microbe metabolites [9]. Prior research has demonstrated a correlation between elevated TMAO levels and HF and coronary artery disease (CAD) [10,11]. Recently, our TAMIR (Impact of TMAO Serum Levels on Hyperemic IMR in STEMI Patients) study (NCT05406297) showed that TMAO levels were associated with CMD after STEMI [12]. Thus, there is a potential pathophysiologic link between TMAO and post-STEMI HF. This study investigated TMAO as a biomarker for predicting FLVR or LVDD following STEMI.

## 2. Results

### 2.1. Study Population

Between January 2021 and July 2022, 624 consecutive STEMI patients underwent PPCI. Of those, 400 had non-culprit lesion(s) and 190 met one of the exclusion criteria. Hence, 210 patients were included. Over the 12-month follow-up period, six patients died. Therefore, 12-month echocardiography data were available for 204 patients (Figure 1). The median age was 65 (IQR 57–76) years and 81 patients (39.7%) were female.

At the 12-month follow-up, 40 (19.7%) of the 204 patients had grade 2 LVDD, with none having grade 3 LVDD. Meanwhile, 148 (72.55%), 24 (11.76%), 17 (8.33%), and 15 (7.35%) patients developed group 1, 2, 3, and 4 FLVR, respectively. Overall, 47 (23.0%) patients were affected with either group 4 FLVR or ≥grade II LVDD, with 32 patients having only LVDD, 7 with group 4 FLVR only, and 8 patients with both LVDD and group 4 FLVR. The baseline and procedure characteristics are outlined in Table 1, while the laboratory results are provided in Table 2. The median TMAO at the baseline was 0.94 (IQR 0.62–1.95) µM; at 3 months was 2.91 (IQR 1.66–4.71) µM; and at the 12-month follow-up was 2.44 (IQR 1.35–4.02) µM. The TMAO levels increased significantly from the baseline to 3 months, but remained stable until 12 months (Figure A1).

The baseline characteristics between affected patients compared to those not affected were similar, other than the female sex (61.70% vs. 34.36%, *p* = 0.001) and diabetes (38.30% vs. 20.25%, *p* = 0.019). There was no between-group difference in medication use at the baseline or 3-month follow-up. Laboratory data, including peak troponin levels and baseline TMAO levels, were similar between groups at the baseline. However, differences emerged at the 3-month follow-up, with significantly higher levels of BNP, hs-CRP, and 3- and 12-month TMAO in the affected group compared with the unaffected group. Notably, the affected group had higher TMAO increases from the baseline to 3 months compared to those in the unaffected group.

### 2.2. Association Between Trimethylamine N-Oxide and LV Systolic and Diastolic Function

The echocardiographic parameters post-PCI and at the 12-month follow-up are shown in Table 3. Both groups had similar baseline echocardiographic results, except for the E/A ratio, which was significantly higher in the affected group (1.3 vs. 0.84, *p* < 0.001). Figure 2 shows the correlations between biomarkers, including TMAO levels at the baseline and 3 months, an increase from the baseline to 3 months, peak troponin, BNP, and hs-CRP, and the 12-month echocardiographic parameters, including LVEF, delta LVEF, LVEDV, E, E/A, E/e′, TRpV, and LAVI. The TMAO level at the baseline was not correlated with any 12-month echocardiographic parameters, while the 3-month TMAO and delta TMAO showed significant correlations with most 12-month echocardiographic parameters except for the 12-month LVEDV. The distribution of the 12-month echocardiographic parameters and quartiles of 3-month TMAO levels is shown in Table A1. Similarly, the 3-month BNP and hs-CRP demonstrated significant associations with most echocardiographic parameters, with the exceptions of the 12-month E/e′ ratio and LAVI for BNP and the 12-month LAVI and changes in LVEF and LVEDV for hs-CRP. In contrast, there were no significant associations between the peak troponin levels and echocardiographic parameters except for at 12 month.

Cubic spline curves analysis revealed a significant association between 3-month TMAO levels and the log odds of developing group 4 FLVR or ≥grade II LVDD. Figure 3A illustrates a linear increase in the log odds of developing group 4 FLVR when the TMAO is above three. Figure 3B shows a sharp increase in the log odds of developing ≥ grade II LVDD when the TMAO was between three and six.

### 2.3. The Diagnostic Performance of TMAO and Biomarkers to Detect Significant FLVR and Significant LV Diastolic Dysfunction

The area under the curve (AUC) of the ROC analysis for using 3-month TMAO to predict group 4 FLVR and ≥grade II LVDD were 0.77 (95% CI 0.63–0.91) and 0.72 (95% CI 0.63–0.81), respectively. The central illustration (Figure 1) illustrates the ROC analysis when using baseline echocardiographic parameters (post-PCI LVEF and E/A ratio) and biomarkers (3-month TMAO, peak troponin, hs-CRP, and BNP) as predictors for group 4 FLVR and ≥grade II LVDD. Overall, the 3-month TMAO, BNP, and post-PCI E/A ratio have a similar diagnostic performance for discriminating group 4 FLVR, whilst the 3-month TMAO and delta TMAO were the only parameters with an AUC > 0.70 for LVDD. A comparison of the performance of all these predictors is shown in Table A2. In the prospectively collected validation cohort (n = 31), the AUC or ROC for using 3-month TMAO to predict group 4 FLVR and ≥grade II LVDD were 0.77 (95% CI 0.56–0.99) and 0.77 (95% CI 0.57–0.97), respectively (Figure A2).

### 2.4. Trimethylamine N-Oxide’s Additional Value in LV Dysfunction Diagnosis

ROC curve analysis was employed to evaluate the diagnostic performance of traditional risk factors both alone and in combination with biomarkers. The AUC for traditional risk factors alone was 0.79 (95% CI: 0.69–0.90) for group 4 FLVR and 0.68 (95% CI: 0.58–0.77) for ≥grade II LVDD. The addition of hs-CRP and BNP increased the AUCs to 0.92 (95% CI: 0.87–0.98) and 0.69 (95% CI: 0.60–0.78) with *p*-values of 0.009 and 0.406, respectively. Incorporating all three biomarkers—TMAO, hs-CRP, and BNP—further enhanced the AUCs to 0.93 (95% CI: 0.88–0.99) and 0.75 (95% CI: 0.66–0.84), with *p*-values of 0.032 and 0.020, respectively (Figure 4A and Figure 4B). The NRI and IDI analysis showed that the addition of hs-CRP and BNP to traditional risk factors did not significantly improve the ability to predict group 4 FLVR and ≥grade II LVDD. However, the addition of TMAO, hs-CRP, and BNP to traditional risk factors significantly improved the prediction of ≥grade II LVDD, but not group 4 FLVR (Table A3).

### 2.5. Multivariable Logistic Regression Analysis

Multivariable logistic regression analysis was used to identify independent risk factors for group 4 FLVR and ≥grade 2 LVDD, which are shown in Table 4 and Table 5. TMAO emerged as an independent predictor for both conditions, with a hazard ratio of 1.28 (95% CI: 1.12–1.47, *p* < 0.001) for group 4 FLVR and 1.29 (95% CI: 1.13–1.50, *p* < 0.001) for ≥grade 2 DD.

### 2.6. Trimethylamine N-Oxide and Clinical Outcomes at 12 Months

At the 12-month follow-up, MACCE occurred in 44 (20.95%) patients. The cumulative event curve showed an increased incidence of MACCE in the highest TMAO quartile (log-rank *p* < 0.001), starting in the first-month post-STEMI (Figure 5), and was driven predominantly by HF hospitalization (Figure A3). The 15 HF hospitalizations predominantly occurred in patients within the highest TMAO quartile, showing a bimodal distribution during the first month and again after three months (Figure A4).

## 3. Discussion

In this single-center prospective study of 204 STEMI patients with multivessel CAD, we investigated the potential role of the novel biomarker TMAO, among other common biomarkers, for identifying patients at high risk of developing FLVR or LVDD despite successful FFR-guided complete revascularization. The main findings of this study are:

1.The TMAO level 3 months after STEMI is significantly associated with poorer recovery in LV systolic function, LVDD, FLVR, and a higher incidence of MACCEs at 12 months.2.The TMAO level at 3 months is one of the best biomarkers for predicting ≥grade II LVDD and group 4 FLVR, outperforming the baseline echocardiographic parameters, and is validated in a prospectively enrolled cohort.3.The addition of TMAO and other biomarkers to traditional risk factors significantly improves the prediction of ≥grade II LVDD and group 4 FLVR.4.In multivariable logistic regression adjusted for clinical factors and other biomarkers, the 3-month TMAO remains an independent predictor of developing ≥grade II LVDD and group 4 FLVR.

Our findings demonstrate that TMAO is a potential biomarker for identifying high-risk patients of developing LV sequelae after STEMI for early treatment.

### 3.1. LV Functional Remodeling, Diastolic Dysfunction, and Adverse Outcomes After STEMI

The impact and recovery of LV function post-STEMI have been the focus of research for nearly half a century [13]. Over time, the incidence of LV remodeling, traditionally defined as a 20% increase in LVEDV, has decreased dramatically primarily due to the universal adoption of PPCI and the widespread use of beta-blockade and angiotensin-converting enzyme inhibitors/angiotensin II blockers [14]. Paradoxically, the incidence of post-STEMI HF has risen as more patients survive the initial event, but then struggle with severely damaged hearts.

Cardiac magnetic resonance imaging (CMR) in patients with LV remodeling reveals extensive cardiomyocyte disorganization, a loss of myocardium sheetlet annularity, edema, and fibrosis [15]. However, recent large-scale studies have shown that in the modern era, the traditional LV remodeling definition is no longer sufficient to distinguish high-risk patients [14]. Thus, Chimed et al. developed the FLVR definition, which integrates the LV systolic function to improve the assessment of prognosis. Notably, they demonstrated that not all LV dilatation (classic LV remodeling) is associated with poor prognosis, with this confined to only those with LV dilatation and functional impairment (group 4 FLVR) [4]. A recent CMR study corroborates these findings [16]; however, given the low incidence of group 4 FLVR, additional surrogates are needed [17].

Impairment in myocardial relaxation is among the earliest reactions to ischemia, as demonstrated in early percutaneous transluminal coronary angioplasty studies [18]. LVDD remains a common sequela after STEMI despite successful PPCI and medical therapy. Yoon et al. showed that despite the recovery in systolic function, in more than half of the patients, the diastolic function did not improve at the 6-month follow-up, with these patients having a higher risk of death, recurrent MI, and HF hospitalization afterward. Furthermore, Bae et al. demonstrated that diastolic function parameters are as independently associated with mortality and HF hospitalization after MI, with a prognostic performance better than LVEF [5]. Thus, both FLVR and LVDD can serve as useful surrogate endpoints in the STEMI population, offering prognostic information beyond traditional LVEF or LV remodeling.

### 3.2. Association Between Trimethylamine N-Oxide and LV Systolic and Diastolic Dysfunction

In HF patients, the gut is the first organ to be ischemic and the last to recover, which, when combined with systemic congestion and gut edema, leads to dysbiosis, chronic inflammation, and increased intestinal permeability [19]. TMAO levels reflect changes in the composition of the gut microbiota and are associated with a worse HF status and outcomes [20]. Multiple animal studies have demonstrated that TMAO can directly induce: (1) inflammation, (2) adverse LV remodeling, (3) reduced LV systolic function, and (4) myocardial fibrosis, all of which collectively contribute to worsening HF [20]. Furthermore, TMAO can indirectly promote HF progression by inducing renal fibrosis, renal tubular dysfunction, and atherosclerosis events such as plaque rupture [21,22].

There have only been a few studies investigating the association between TMAO and LV remodeling or LVDD. Tang et al. evaluated the association between TMAO and LV function in 112 chronic HF patients. They reported a positive correlation between the TMAO level and diastolic function parameters as well as worse outcomes; however, no significant association between TMAO and the parameters of LV remodeling, namely LVEDV and LVEF, was seen [23]. Of note, this study only included patients with significant HF (LVEF ≤ 35%), potentially limiting further stratification. Conversely, Organ et al. demonstrated in a murine HF model that withdrawing dietary TMAO or inhibiting its formation with a choline trimethylamine lyase inhibitor can significantly attenuate adverse LV remodeling, suggesting a causal relationship [24].

To the best of our knowledge, our study is the first to report the association between TMAO levels and FLVR and LVDD in the post-STEMI population. These findings provide insights into the potential mechanism behind the development of HF in this population and offer a novel approach for early stratification and treatment.

This study interestingly revealed that 3-month TMAO levels, rather than baseline TMAO, were linked to poor LV recovery and clinical outcomes. We believe the causes are multiple. 1. In our cohort, the baseline TMAO levels were low, aligning with the findings of Almesned et al. and Matsuzawa et al., likely due to fasting, vomiting, and dehydration during acute illness status. 2. Notably, all STEMI patients were given aspirin-loading doses and statin, which were known to reduce plasma TMAO levels. 3. The patient’s microbiota and metabolism may continue to evolve after the acute event, as shown in previous studies in STEMI patients [25,26,27,28,29]. Therefore, the 3-month TMAO may more accurately represent the patients’ baseline metabolic status and diet and, consequently, their clinical outcomes.

### 3.3. Deciphering the Pathway of Trimethylamine N-Oxide and Potential Interventions

With established evidence from animal models and HF cohorts, the gut-TMAO-HF axis may be a potential therapeutic target to reduce the incidence of post-STEMI HF. The BIOSTAT-CHF (biological research on the personalized treatment of chronic HF) study revealed that while the BNP levels responded well to current guideline-directed medical therapy, there was no impact on the TMAO levels. This suggests that TMAO impacts HF prognosis through alternative pathways not controlled by current medical therapy. Importantly, in our study, medication use did not differ between the affected group and the unaffected, suggesting room for further medical and dietary intervention. In humans, dietary choline, carnitine, and butyrobetaine are transformed by certain types of gut microbes, mainly Firmicutes and Proteobacteria, to trimethylamine, which is then metabolized in the liver by human FMO 1 and 3 into TMAO [30]. Several therapeutic options have been developed to reduce circulating TMAO levels.

Dietary modifications: The Mediterranean diet has been reported to reduce TMAO levels and the incidence of heart failure in large studies [31,32]. Similarly, a high-fiber diet or DASH diet could prevent the development of heart failure and improve cardiac remodeling [33]. Probiotics: certain strains have demonstrated cardioprotective effects by reducing LV hypertrophy, improving cardiac function, and decreasing inflammatory markers [34]. Prebiotics, such as inulin, can promote the growth of beneficial bacteria, improve insulin sensitivity, and reduce systemic inflammatory markers in obese patients [35]. Microbial TMA-lyase inhibitors: compounds such as fluoromethylcholine (FMC), iodomethylcholine (IMC), and 3,3-Dimethyl-1-butanol (DMB) reduce TMAO levels in animal models without affecting the viability of the symbiotic bacteria [36,37]. Natural phytochemicals such as allicin, resveratrol, indole-3-carbinol, berberine, and 3,3′-diindolylmethane could potentially reduce TMAO formation, but the potential adverse effects necessitate the need for careful clinical studies [20]. Fecal microbial transplantation (FMT) from lean donors has shown promise in improving the metabolic status of patients but has never been tested in the heart failure population [38]. Lastly, short-term antibiotic therapy, such as orally administered vancomycin, has been shown to reduce the infarct size in animal models and demonstrated the inhibition of TMAO synthesis in humans [39,40]. However, this protective effect must be weighed against its potential toxicity and the risk of drug-resistant microbiota. These strategies offer novel treatment avenues for heart failure, although further research is warranted to fully elucidate their mechanism and to optimize their clinical application.

### 3.4. The Role of Traditional Risk Factors and Other Biomarkers in FLVR and LVDD

As demonstrated in our study, after STEMI, the functional and morphological LV sequala remains common despite PPCI, highlighting the crucial role of clinical risk factors and biomarkers in the early identification of high-risk patients [13]. The literature has consistently shown that advanced age and the female sex are the strongest predictors of HF, likely due to the higher prevalence of cardiac hypertrophy in the elderly and the disproportionate impact of comorbidities on women. In addition, the initial infarct size, hypertension, and chronic kidney disease (CKD) have been reported as risk factors for post-STEMI HF [41].

Beyond the potential impact of the microbiota as described above, both LV remodeling and LVDD involve complex pathophysiological pathways, including microvascular injury, myocardial hypertrophy, myocardial fibrosis, inflammation, neurohormonal activation, and the cardio–renal interaction [41]. However, common biomarkers for the infarct size (e.g., troponin level), inflammation (e.g., Hs-CRP), and ventricular extension (BNP) have shown inconsistent results in predicting LV remodeling [42]. Notably, studies showed that the combined assessment of multiple biomarkers provides incremental prognostic information for LV remodeling [7]. Our study corroborates these findings, showing that the combination of the three most promising biomarkers with clinical risk factors significantly improves the prediction model. These results can be easily validated in other post-STEMI cohorts, potentially leading to the development of a clinically useful prediction score.

### 3.5. Limitations

Our study has several limitations: First, the single-center design and focus on patients with multivessel CAD limits the generalizability. However, since data regarding LV remodeling and recovery in the complete revascularization era remain limited, our data provided a piece of timely information for the community. Second, the absence of detailed dietary data and microbiota profiles precludes an analysis of the dietary effects on TMAO and the gut microbiota. Food rich in choline and carnitine, such as red meat, eggs, and fish, has been shown to elevate TMAO concentrations [43,44]. Additionally, genetic variation in the hepatic enzyme flavin-containing monooxygenase 3 (FMO3), which plays a crucial role in converting dietary-derived trimethylamine into TMAO, may further influence the systemic TMAO levels [45]. However, our study population is relatively homogeneous, comprising exclusively white Lithuanian individuals, which may reduce the dietary variability. Third, the observational nature of this study limits our ability to establish causality between TMAO levels and FLVR or LVDD. While the animal studies described above support a causal relationship between TMAO exposure and HF, these models were not specific to MI. Third, our primary focus on TMAO may not fully capture the role of other emerging biomarkers. However, the gut-TMAO-HF axis offers unique therapeutic options that could be explored. Moreover, the 12-month follow-up period may not fully reveal the long-term impact of TMAO on LV function. Lastly, we could not perform follow-up CMR, limiting more sophisticated analyses. Thus, future multi-center studies with extended follow-ups, broader inclusion, CMR, and a comprehensive exploration of the gut microbiota are essential to validate and extend our results.

## 4. Materials and Methods

### 4.1. Study Design

This is a sub-study of the prospective, observational TAMIR study (NCT05406297) conducted at the Lithuanian University of Health Sciences Kaunas Clinics, Kaunas, Lithuania. This study enrolled STEMI patients with multivessel CAD who underwent successful PPCI of the culprit lesion. At 3 months post PPCI, all patients underwent staged PCI for fractional flow reserve (FFR)-guided complete revascularization. Additionally, an invasive physiological assessment of CMD was performed in the culprit arteries, with the results described in the TAMIR study [12]. To cross-validate the results of this study, we prospectively enrolled 31 additional patients using the same inclusion and exclusion criteria.

### 4.2. Inclusion and Exclusion Criteria

This study enrolled patients ≥ 40 years old with STEMI and multivessel CAD, defined as having >50% stenosis in two or more major coronary arteries of a 2.5 mm diameter or more. All patients received a loading dose of dual antiplatelet therapy (300 mg of acetylsalicylic acid and 180 mg of ticagrelor or 600 mg of clopidogrel) at least 30 min prior to the PPCI. Patients with a history of acute coronary syndrome before the index procedure, those who received fibrinolysis prior to the PPCI, or those with moderate or severe stenosis or insufficiency of the aortic or mitral valve were excluded. Individuals with chronic infections or severe liver or kidney dysfunction, potentially affecting their gut microbiota, were also excluded. Additionally, patients who received coronary artery bypass grafting after the PPCI were also excluded.

### 4.3. Data Collection and Echocardiographic Analysis

Clinical data, including demographics, medical history, procedure details, laboratory results, and medication regimens, were collected prospectively. Venous blood samples were taken within 24 h of the PPCI (baseline) and at the 3-month and 12-month follow-ups. The plasma TMAO levels were quantified using the TMAO ELISA Kit from the Bioassay Technology Laboratory, Shanghai, China. This TMAO ELISA kit has an intra-assay and inter-assay precision of coefficient of variation <8% and <10%, respectively. For quantification, a standard calibration curve was generated using serial dilutions of known TMAO concentrations provided in the kit. Absorbance was measured at the recommended wavelength using the Infinite^®^ M Plex microplate reader, and plasma TMAO concentrations were determined by interpolating the sample optical density values against the standard curve. All samples and standards were run in duplicate to ensure analytical precision, and the results were expressed in µmol/L (µM). Other biomarkers, including brain natriuretic peptide (BNP) and high-sensitivity C-reactive protein (hs-CRP), were sampled at the 3-month follow-up visits. Certified echocardiographers, blinded to the study data, conducted an echocardiography 24 h after STEMI and at the 1-year follow-up with the same EPIQ 7 ultrasound system (Washington, DC, USA: Phillips Ultrasound, Inc.). The measurement of the LVEF and left atrial (LA) volume was conducted in accordance with the European Association of Cardiovascular Imaging (EACVI) guidelines [46]. The LV diastolic function was assessed following the American Society of Echocardiography 2016 guidelines [47].

### 4.4. Study Endpoints and Definitions

The primary endpoints of this study were (1) the incidence of group 4 FLVR or (2) grade II or grade III LVDD at twelve months after STEMI.

FLVR was defined according to Chimed et al.’s classification as Group 1 patients with no significant change in the LV diameter and LVEF; Group 2 patients with no LV dilatation but an absolute reduction in LVEF of >5%; Group 3 patients with an increase in the LV end-diastolic volume (LVEDV) ≥ 20% but no impairment in LVEF; and Group 4 patients with both LVEDV dilatation ≥ 20% and an absolute reduction in LVEF of >5% [4].

The secondary endpoint was the incidence of major adverse cardiovascular and cerebrovascular events (MACCE), defined as the patient-oriented composite endpoint (POCE) plus hospitalizations for HF. POCE was defined as the composite of all-cause death, any myocardial infarction, any revascularization, and both ischemic and hemorrhagic strokes [48].

### 4.5. Statistical Analysis

Continuous variables were presented as means ± SD or medians with interquartile ranges (IQR) and were compared with the Student’s *t*-test or Mann–Whitney *U* test as appropriate. Categorical variables were presented as frequencies and percentages and compared with Chi-Square or Fisher’s exact tests as appropriate. Spearman’s linear correlation analysis was conducted to examine the relationship between biomarkers, including the peak troponin level, TMAO, BNP, and hs-CRP, and echocardiography parameters at the 12-month follow-up, which included LVEF, LVEDV, E/e’, TRpV, LAVI, and the change in LVEF and LVEDV from the baseline.

Receiver Operating Characteristic (ROC) curve analysis with the Delong test and Youden index was used to compare its diagnostic performance and identify the optimal cut-offs for detecting group 4 FLVR and ≥grade II LVDD, respectively. The improvements in the discrimination capacity between using traditional risk factors alone (age, sex, hypertension, diabetes, hyperlipidemia, and smoking) or in combination with BNP and CRP (model 1) and lastly with the addition of TMAO (model 2) were evaluated with the Net Reclassification Improvement (NRI) and the Integrated Discrimination Improvement (IDI) analysis. Logistic regression models were used to calculate the odds ratios for group 4 FLVR and ≥grade II LVDD per unit increase of TMAO. Multivariable models were adjusted for the effect of the relevant patient demographics, medical history, clinical course, lab values, and angiographic characteristics (*p* < 0.05 for inclusion). Kaplan–Meier analysis was used to assess event-free survival and was compared with log-rank tests. Restricted cubic spline curves were created to explore the dose-response of the TMAO level with group 4 FLVR and ≥grade II LVDD. Statistical significance was set at *p* < 0.05. Data were processed using R Studio (Build 576).

### 4.6. Ethics Approval and Consent to Participate

This study complies with the Declaration of Helsinki and was approved by the Kaunas Regional Biomedical Research Ethics Committee (nr: BE-2-5). All participants or their legal representatives provided written informed consent prior to this study. This study was registered in the clinical trials registry with the identifier NCT05406297.

## 5. Conclusions

The novel biomarker TMAO is independently associated with LVDD and FLVR after STEMI, with a superior performance compared to echocardiographic parameters and other biomarkers, and could improve the prediction of traditional risk factors. These results suggest that monitoring biomarkers could be a cost-effective strategy for identifying and pre-emptively treating high-risk STEMI patients.

## Figures and Tables

**Figure 1 ijms-26-03400-f001:**
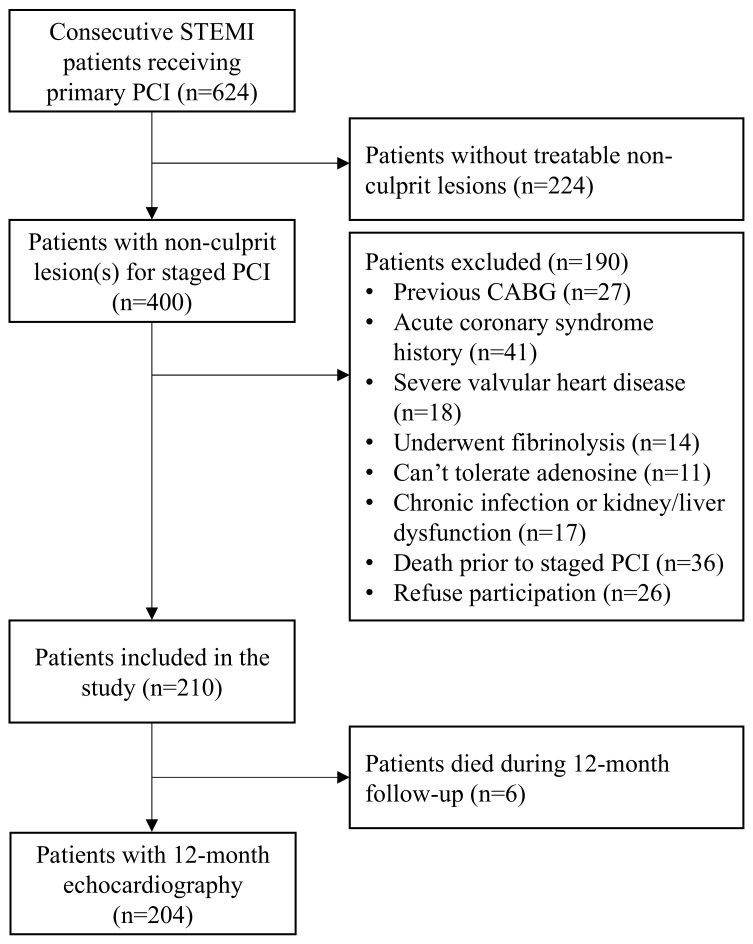
Study flow chart. Selection process and patient enrolment flow chart. STEMI = ST-segment elevation myocardial infarction, PCI = percutaneous coronary intervention, CABG = coronary artery bypass graft surgery.

**Figure 2 ijms-26-03400-f002:**
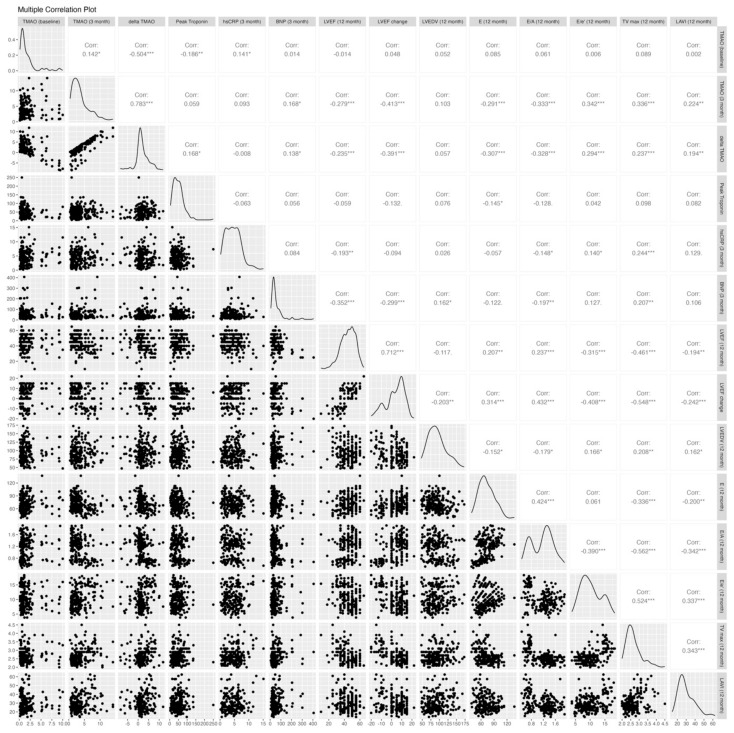
Correlation analysis of TMAO levels and echocardiographic parameters. E/A, early-to-late diastolic filling velocity; average e’, averaged diastolic myocardium velocity at the septal and lateral side of the mitral annulus; E/e’, the ratio between early mitral inflow velocity and mitral annular early diastolic velocity; LV, left ventricular; LVEDV, LV end-diastolic volume; LVEF, LV ejection fraction; LAVI, left atrium volume index; and TRpV, tricuspid regurgitation peak velocity. * *p* ≤ 0.05; ** *p* ≤ 0.01; *** *p* ≤ 0.001.

**Figure 3 ijms-26-03400-f003:**
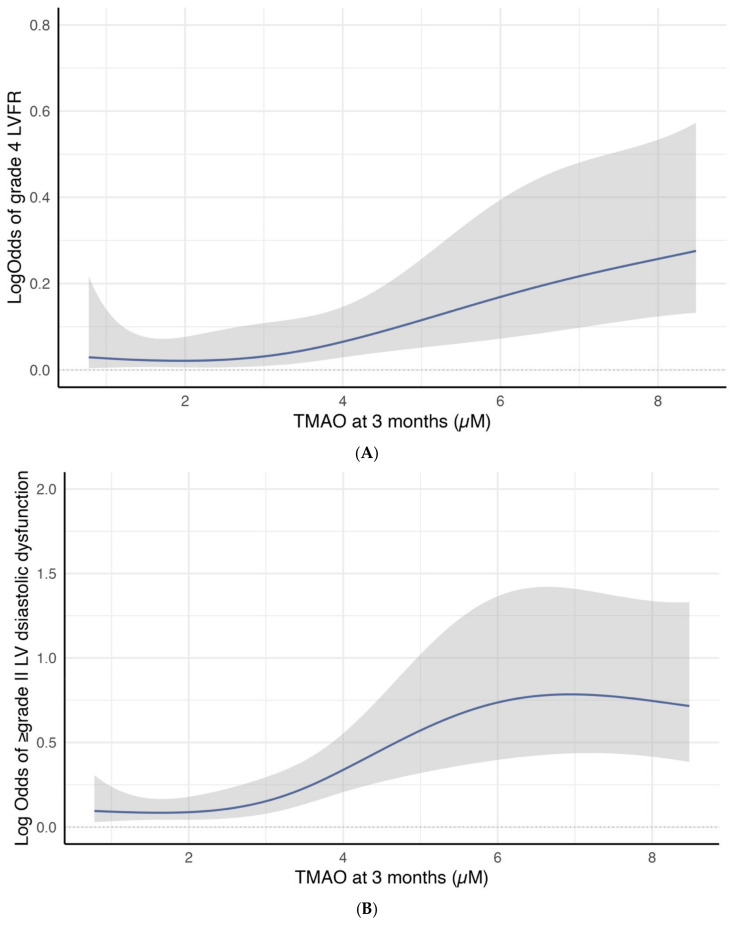
(**A**) Restricted cubic spline curve for the association between 3-month TMAO level and the log odds of group 4 FLVR at 12 months. (**B**) Restricted cubic spline curve for the association between 3-month TMAO level and the log odds of grade II or III DD at 12 months.

**Scheme 1 ijms-26-03400-sch001:**
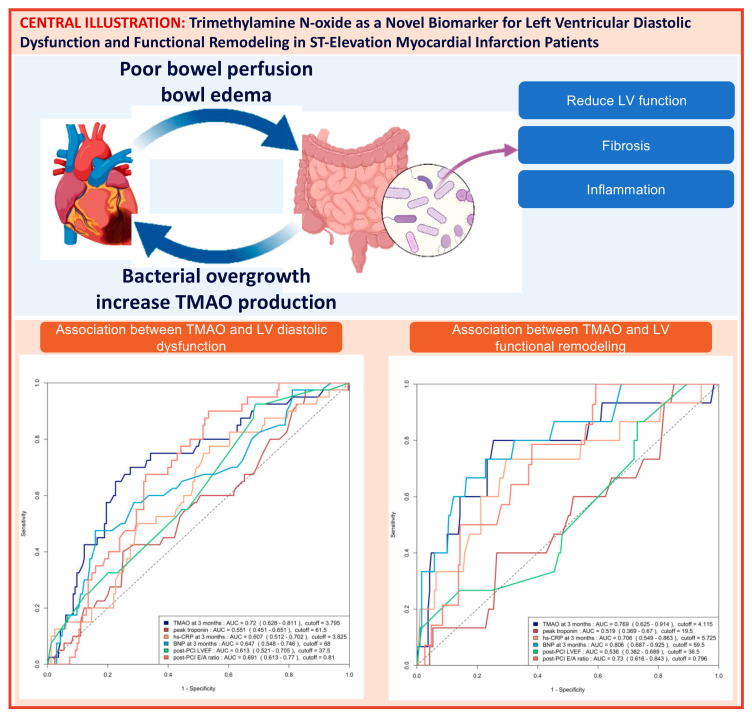
Central illustration: Heart failure and gut dysbiosis can mutually potentiate each other (**left upper panel**). The mechanism with which Trimethylamine N-oxide (TMAO) induces cardiac dysfunction was demonstrated in the **right upper panel**. The ROC curve analysis of predicting ≥ grade II left ventricular diastolic dysfunction (LVDD) with baseline echocardiographic parameters and biomarkers and the ROC curve analysis of predicting group 4 functional left ventricular remodeling (FLVR) with baseline echocardiographic parameters and biomarkers are shown in the **lower left** and the **lower right panel**, respectively. AUC = area under the ROC curve, CRP = C-reactive protein, BNP = brain natriuretic peptide.

**Figure 4 ijms-26-03400-f004:**
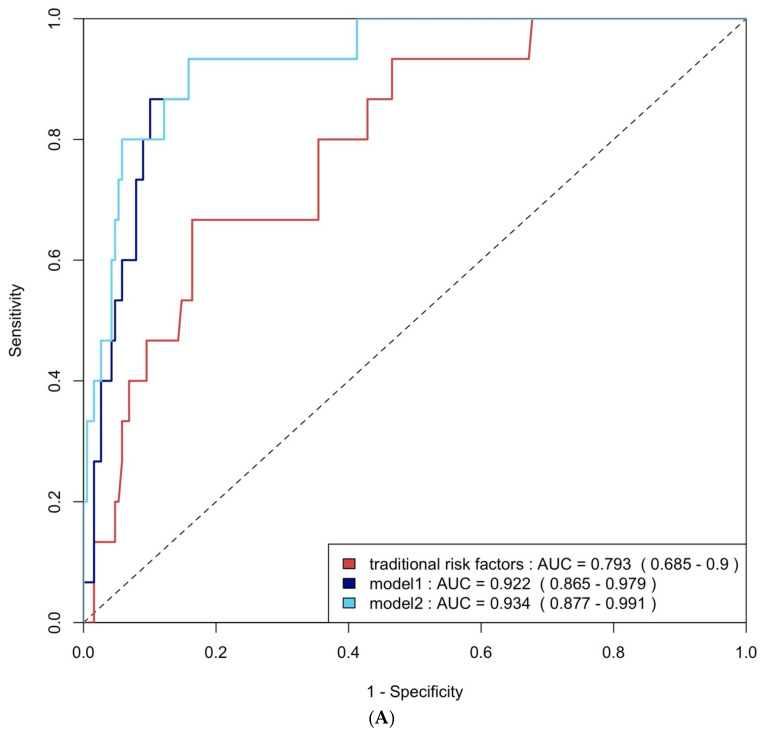
(**A**) ROC curve analysis of predicting group 4 FLVR with traditional risk factors and models with biomarkers. (**B**) ROC curve analysis of predicting grade II or III DD with traditional risk factors and models with biomarkers. Traditional risk factors: age, gender, hypertension, diabetes, hyperlipidemia, and smoking. Model 1: traditional risk factors + hs-CRP and BNP level at 3 months, Model 2: Model 1 + 3 months of TMAO.

**Figure 5 ijms-26-03400-f005:**
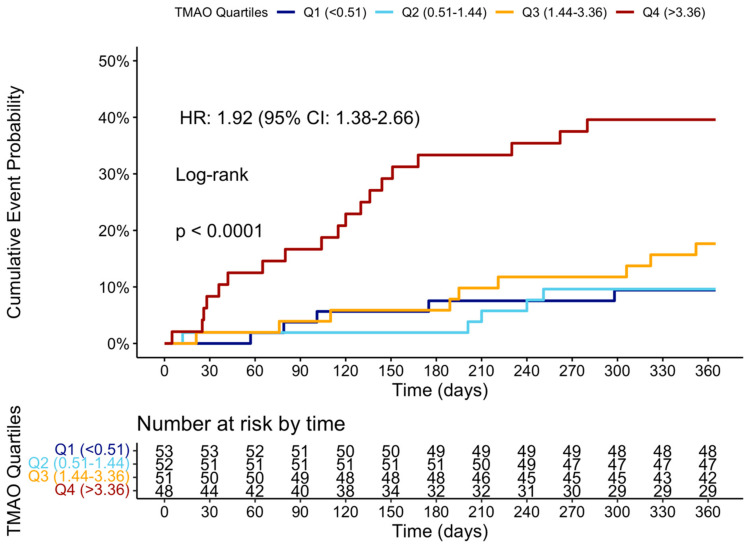
Cumulative event curve of TMAO quartiles and incidence of MACCE at 1 year. MACCE = major adverse cerebral and cardiovascular events.

**Table 1 ijms-26-03400-t001:** Baseline and index procedure characteristics.

Baseline Characteristic	Overall (n = 204)	No Primary Endpoint (n = 157)	Group 4 FLVR or Grade II Diastolic Dysfunction (n = 47)	*p*-Value
Sex (Female)	81 (39.7)	52 (33.1)	29 (61.7)	0.001
Age (years)	65.00 [57.00, 76.00]	66.00 [57.00, 76.00]	63.00 [57.50, 76.00]	0.550
Body mass index (kg/m^2^)	27.46 [24.60, 30.68]	27.44 [24.77, 31.02]	27.68 [24.23, 29.64]	0.366
Body surface area (m^2^)	1.93 [1.81, 2.10]	1.93 [1.79, 2.09]	1.97 [1.85, 2.11]	0.460
Primary Diagnosis				0.999
Anterior STEMI	115 (56.37%)	88 (56.05%)	27 (57.45%)	
Inferior STEMI	115 (56.37%)	69 (43.95%)	20 (42.55%)	
Arterial hypertension	120 (58.8)	92 (58.6)	28 (59.6)	1
History of coronary artery disease	55 (27.0)	41 (26.1)	14 (29.8)	0.756
History of PCI	24 (11.8)	18 (11.5)	6 (12.8)	1
History of stroke	27 (13.2)	17 (10.8)	10 (21.3)	0.108
History of diabetes mellitus	49 (24.0)	31 (19.7)	18 (38.3)	0.016
History of dyslipidemia	117 (57.4)	90 (57.3)	27 (57.4)	1
Smoker (former/current)	106 (52.0)	78 (49.7)	28 (59.6)	0.306
History of alcohol abuse	19 (9.3)	15 (9.6)	4 (8.5)	1
KILLIP class				0.810
I	61 (29.9)	47 (29.9)	14 (29.8)	
II	109 (53.4)	82 (52.2)	27 (57.4)	
III	26 (12.7)	22 (14.0)	4 (8.5)	
IV	8 (3.9)	6 (3.8)	2 (4.3)	
Baseline medications				
Beta-blockers	32 (15.7)	22 (14.0)	10 (21.3)	0.331
Calcium channel blocker	62 (30.5)	45 (28.8)	17 (36.2)	0.438
ACEI/ARB	40 (19.6)	32 (20.4)	8 (17.0)	0.764
Aldosterone antagonists	2 (1.0)	1 (0.6)	1 (2.1)	0.947
Statins	35 (17.2)	28 (17.8)	7 (14.9)	0.804
Aspirin	42 (20.7)	29 (18.6)	13 (27.7)	0.254
3-month follow-up medications				
Beta-blockers	176 (86.3)	134 (85.4)	42 (89.4)	0.646
Calcium channel blocker	194 (95.1)	147 (93.6)	47 (100.0)	0.165
ACEI/ARB	11 (5.4)	8 (5.1)	3 (6.4)	1
Aldosterone antagonists	39 (19.1)	28 (17.8)	11 (23.4)	0.522
Statins	197 (96.6)	150 (95.5)	47 (100.0)	0.309
Aspirin	204 (100.0)	157 (100.0)	47 (100.0)	1
Ticagrelor	177 (86.8)	138 (87.9)	39 (83.0)	0.530
Clopidogrel	29 (14.2)	20 (12.7)	9 (19.1)	0.387
Procedure Characteristics				
Pain-to-door time (minutes)	314.00 [109.25, 597.00]	322.00 [104.00, 597.00]	262.00 [134.00, 502.50]	0.884
Door-to-balloon (minutes)	40.00 [29.75, 51.25]	40.00 [29.00, 51.00]	40.00 [32.50, 55.00]	0.419
Pre-PCI TIMI flow				0.220
0	126 (61.8)	91 (58.0)	35 (74.5)	
1	7 (3.4)	6 (3.8)	1 (2.1)	
2	44 (21.6)	36 (22.9)	8 (17.0)	
3	27 (13.2)	24 (15.3)	3 (6.4)	
Post-PCI TIMI flow				0.073
0	2 (1.0)	1 (0.6)	1 (2.1)	
1	1 (0.5)	1 (0.6)	0 (0.0)	
2	22 (10.8)	13 (8.3)	9 (19.1)	
3	179 (87.7)	142 (90.4)	37 (78.7)	
Culprit Vessel Territory				0.482
LAD	117 (57.4)	89 (56.7)	28 (59.6)	
LCX	47 (23.0)	39 (24.8)	8 (17.0)	
RCA	40 (19.6)	29 (18.5)	11 (23.4)	
Number of diseased vessels				0.916
2-Vessel Disease	118 (57.84%)	90 (57.32%)	28 (59.57%)	
3-Vessel Disease	86 (42.2)	67 (42.7)	19 (40.4)	

Values are presented as n (%) and median [1st quartile, 3rd quartile] for categorical variables and numeric variables, respectively. STEMI = ST elevation myocardial infarction; PCI = Percutaneous Coronary Intervention; LAD = left anterior descending artery; LCX = circumflex artery; RCA = right coronary artery.

**Table 2 ijms-26-03400-t002:** Laboratory parameters.

Parameters	Overall (n =204)	No Primary Endpoint (n = 157)	Group 4 FLVR or Grade II Diastolic Dysfunction (n =47)	*p*-Value
Baseline laboratory test				
Hemoglobin (g/L)	134.56 ± 20.48	133.93 ± 19.94	136.66 ± 22.27	0.453
White Blood Cell Count (10^9^/L)	9.86 [8.28, 12.09]	9.77 [8.20, 11.97]	10.18 [8.40, 13.26]	0.350
Platelets (×10^9^/L)	240.50 [203.75, 271.50]	238.00 [204.00, 265.00]	255.00 [202.00, 300.00]	0.121
Total cholesterol (mmol/L)	4.67 [3.83, 5.80]	4.69 [3.79, 5.81]	4.56 [3.85, 5.67]	0.869
Low-density lipoprotein (mmol/L)	2.94 [2.13, 3.93]	2.94 [2.16, 3.92]	3.00 [1.87, 4.07]	0.910
High-density lipoprotein (mmol/L)	1.12 [0.95, 1.35]	1.12 [0.95, 1.35]	1.13 [0.94, 1.35]	0.901
Triglycerides (mmol/L)	1.17 [0.83, 1.65]	1.17 [0.82, 1.64]	1.14 [0.84, 1.77]	0.642
Creatinine Clearance (ml/min)	39.40 [34.88, 47.50]	40.20 [35.10, 48.20]	37.40 [34.15, 45.25]	0.125
Baseline Troponin I (µg/L)	2.21 [0.84, 3.71]	2.22 [0.77, 3.87]	2.19 [0.89, 3.01]	0.890
Peak Troponin I (µg/L)	47.00 [27.75, 64.25]	44.00 [27.00, 61.00]	50.00 [29.00, 67.50]	0.333
High-sensitivity CRP (mg/L)	3.73 [1.82, 10.51]	4.14 [2.02, 11.14]	2.81 [1.69, 6.72]	0.181
Baseline TMAO	0.94 [0.62, 1.95]	0.94 [0.62, 1.87]	1.12 [0.45, 1.98]	0.627
3-month follow-up laboratory test				
3-month TMAO (µM)	2.91 [1.66, 4.71]	2.47 [1.36, 3.85]	4.28 [2.56, 7.43]	<0.001
TMAO increase from baseline to 3 months	1.44 [0.51, 3.36]	1.34 [0.47, 2.40]	3.72 [1.39, 6.00]	<0.001
3-month hs-CRP (mg/L)	4.25 [2.31, 6.32]	3.98 [1.98, 6.14]	5.74 [3.91, 6.78]	0.006
3-month BNP (ng/L)	37.00 [27.00, 64.25]	36.00 [27.00, 51.00]	69.00 [32.50, 88.00]	<0.001
12-month follow-up laboratory test				
12-month TMAO (µM)	2.44 [1.35, 4.02]	2.14 [1.18, 3.14]	5.74 [3.00, 8.02]	<0.001

CMD = coronary microvascular dysfunction; BNP = brain natriuretic peptide; hs-CRP = high-sensitivity C-reactive protein; ESR = erythrocyte sedimentation rate; TMAO = Trimethylamine N-oxide; PCI = Percutaneous Coronary Intervention. Values are n (%) or median [IQR].

**Table 3 ijms-26-03400-t003:** Echocardiographic parameters of patients with ST-elevation myocardial infarction.

Parameters	Overall (n = 204)	No Primary Endpoint (n = 157)	Group 4 FLVR or Grade II Diastolic Dysfunction (n = 47)	*p*-Value
*(Post-PCI)*				
LVEF (%)	40.00 [36.00, 45.25]	40.00 [35.00, 45.00]	44.00 [40.00, 48.00]	0.096
LVEDV (mL)	85.40 [68.80, 107.17]	85.20 [68.80, 107.60]	86.60 [69.50, 103.60]	0.729
E/A ratio	0.93 [0.70, 1.37]	0.84 [0.68, 1.32]	1.30 [1.00, 1.53]	<0.001
average e’	8.00 [6.00, 10.00]	8.00 [6.00, 10.00]	8.00 [7.00, 10.00]	0.424
E/e’	8.70 [7.00, 10.77]	8.50 [6.80, 10.67]	9.38 [8.05, 11.08]	0.112
TRpV	2.59 [2.38, 2.80]	2.60 [2.40, 2.80]	2.50 [2.30, 2.75]	0.282
LA volume index	26.07 [22.67, 31.18]	26.17 [22.77, 30.47]	26.01 [22.25, 32.27]	0.816
LV diastolic dysfunction				0.286
Grade 1	190 (93.1)	146 (93.0)	44 (93.6)	
Grade 2	7 (3.4)	6 (3.8)	1 (2.1)	
Grade 3	6 (2.9)	5 (3.2)	1 (2.1)	
NA	1 (0.5)	0 (0.0)	1 (2.1)	
*(at 12-month follow-up)*				
LVEF (%)	45.00 [40.00, 50.00]	48.00 [40.00, 55.00]	36.00 [30.50, 43.50]	<0.001
LVEDV (mL)	89.05 [70.88, 107.75]	85.70 [70.50, 100.80]	97.60 [83.60, 119.80]	0.012
E/A ratio	1.23 [0.77, 1.41]	1.29 [1.12, 1.44]	0.74 [0.69, 0.78]	<0.001
average e’	8.00 [5.00, 10.00]	9.00 [7.00, 10.00]	4.00 [4.00, 5.00]	<0.001
E/e’	9.78 [7.74, 12.65]	8.64 [7.10, 10.62]	15.18 [14.41, 16.00]	<0.001
TRpV	2.60 [2.40, 2.90]	2.46 [2.30, 2.60]	3.10 [2.90, 3.45]	<0.001
LA volume index (mL/m^2^)	27.19 [23.02, 35.83]	26.78 [22.77, 31.38]	37.92 [25.59, 43.68]	<0.001
LV diastolic dysfunction				<0.001
Grade 1	163 (79.9)	157 (100.0)	6 (12.8)	
Grade 2	40 (19.6)	0 (0.0)	40 (85.1)	
Grade 3	0 (0)	0 (0)	0 (0)	
LVEF change (%)	5.00 [0.00, 10.00]	8.00 [0.00, 10.00]	−10.00 [−13.50, 0.00]	<0.001
LVEDV change	0.03 [−0.07, 0.11]	−0.01 [−0.07, 0.08]	0.14 [0.05, 0.31]	<0.001
FLVR				<0.001
Group 1	148 (72.5)	135 (86.0)	13 (27.7)	
Group 2	24 (11.8)	10 (6.4)	14 (29.8)	
Group 3	17 (8.3)	12 (7.6)	5 (10.6)	
Group 4	15 (7.4)	0 (0.0)	15 (31.9)	

Values are presented as n (%) and median [1st quartile, 3rd quartile] for categorical variables and numeric variables, respectively. E/A, early-to-late diastolic filling velocity; average e’, averaged diastolic myocardium velocity at the septal and lateral side of the mitral annulus; E/e’, the ratio between early mitral inflow velocity and mitral annular early diastolic velocity; FLVR, functional LV remodeling; LA, left atrial; LV, left ventricular; LVEDV, LV end-diastolic volume; LVEF, LV ejection fraction; PCI, Percutaneous Coronary Intervention; and TRpV, tricuspid regurgitation peak velocity.

**Table 4 ijms-26-03400-t004:** Multivariable binary logistic analysis for the prediction of group 4 FLVR at 12 months.

Variable	Univariable		Multivariable	
	Odds Ratio	*p*-Value	Odds Ratio	*p*-Value
Age (years)	1.00 (0.96–1.05)	0.909		
Sex (Female)	4.67 (1.43–15.24)	0.011	2.46 (1.12–5.54)	0.027
Body mass index (kg/m^2^)	0.98 (0.88–1.09)	0.696		
Arterial hypertension	1.44 (0.47–4.37)	0.523		
History of coronary artery disease	0.39 (0.09–1.81)	0.231		
History of PCI	Not converge	0.990		
History of stroke	1.72 (0.45–6.54)	0.427		
History of diabetes mellitus	5.59 (1.88–16.63)	0.002	1.31 (0.54–3.04)	0.537
History of dyslipidemia	1.12 (0.38–3.29)	0.830		
Smoker (former/current)	1.06 (0.37–3.04)	0.912		
History of alcohol abuse	1.56 (0.32–7.48)	0.581		
KILLIP class	0.53 (0.18–1.59)	0.256		
Hemoglobin (g/L)	0.99 (0.97–1.02)	0.509		
White Blood Cell Count (10^9^/L)	0.95 (0.79–1.15)	0.595		
Platelets (×10^9^/L)	1.01 (1.00–1.01)	0.088		
Total cholesterol (mmol/L)	1.26 (0.91–1.76)	0.165		
Low-density lipoprotein (mmol/L)	1.42 (0.98–2.07)	0.066		
High-density lipoprotein (mmol/L)	0.90 (0.19–4.34)	0.900		
Triglycerides (mmol/L)	0.74 (0.35–1.58)	0.440		
Creatinine Clearance (mL/min)	0.96 (0.91–1.02)	0.227		
Baseline Troponin I (µg/L)	1.00 (0.91–1.10)	0.929		
Peak Troponin I (µg/L)	1.00 (0.98–1.02)	0.878		
Baseline High-sensitivity CRP (mg/L)	1.00 (0.98–1.02)	0.893		
3-month High-sensitivity CRP (mg/L)	1.32 (1.10–1.59)	0.003	1.12 (0.97–1.3)	0.13
3-month BNP (ng/L)	1.02 (1.01–1.02)	<0.001	0.874	0.874
3-month TMAO	1.34 (1.14–1.58)	<0.001	1.28 (1.12–1.47)	<0.001
Post-PCI LVEF (%)	1.04 (0.96–1.13)	0.32		
Post-PCI LVEDV	1.00 (0.98–1.02)	0.853		
Post-PCI LAVI (ml/m^2^)	1.02 (0.94–1.11)	0.635		
Post-PCI E/A ratio	4.39 (1.36–14.12)	0.013	3.82 (1.55–9.76)	0.004
Post-PCI E/e’	1.03 (0.86–1.24)	0.738		
Post-PCI TRpV	1.26 (0.29–5.41)	0.76		
Pain-to-door time (minutes)	1.00 (1.00–1.00)	0.398		
Door-to-balloon (minutes)	1.00 (0.97–1.03)	0.805		
Culprit Vessel	0.60 (0.16–2.22)	0.441		
Number of diseased vessels	0.47 (0.15–1.54)	0.216		
Stent diameter (millimeters)	2.10 (0.77–5.69)	0.146		
Stent length (millimeters)	1.08 (0.97–1.20)	0.152		

PCI = Percutaneous Coronary Intervention, CRP = C-reactive protein, BNP = brain natriuretic peptide, LVEF = left ventricular ejection fraction.

**Table 5 ijms-26-03400-t005:** Multivariable binary logistic analysis for the prediction of grade II diastolic dysfunction at 12 months.

Variable	Univariable		Multivariable	
	Odds Ratio	*p*-Value	Odds Ratio	*p*-Value
Age (years)	0.98 (0.96–1.01)	0.275		
Sex (Female)	2.87 (1.41–5.83)	0.004	2.46 (1.12–5.54)	0.027
Body mass index (kg/m^2^)	0.96 (0.89–1.03)	0.255		
Arterial hypertension	1.07 (0.53–2.17)	0.843		
History of coronary artery disease	1.60 (0.76–3.36)	0.212		
History of PCI	1.42 (0.52–3.85)	0.489		
History of stroke	2.34 (0.96–5.69)	0.061		
History of diabetes mellitus	1.76 (0.82–3.77)	0.145	1.31 (0.54–3.04)	0.537
History of dyslipidemia	0.90 (0.45–1.80)	0.760		
Smoker (former/current)	1.34 (0.66–2.69)	0.415		
History of alcohol abuse	0.45 (0.10–2.04)	0.302		
KILLIP class	1.30 (0.59–2.88)	0.519		
Hemoglobin (g/L)	1.01 (0.99–1.03)	0.171		
White Blood Cell Count (10^9^/L)	1.13 (1.01–1.27)	0.032		
Platelets (×10^9^/L)	1.01 (1.00–1.01)	0.03	1.01 (1.00–1.01)	0.096
Total cholesterol (mmol/L)	0.92 (0.72–1.16)	0.477		
Low-density lipoprotein (mmol/L)	0.96 (0.73–1.25)	0.747		
High-density lipoprotein (mmol/L)	0.62 (0.21–1.83)	0.389		
Triglycerides (mmol/L)	0.84 (0.55–1.29)	0.425		
Creatinine Clearance (mL/min)	0.96 (0.93–1.00)	0.056		
Baseline Troponin I (µg/L)	0.98 (0.91–1.06)	0.626		
Peak Troponin I (µg/L)	1.00 (0.99–1.01)	0.523		
Baseline High-sensitivity CRP (mg/L)	0.99 (0.97–1.01)	0.284		
3-month High-sensitivity CRP (mg/L)	1.17 (1.03–1.34)	0.017	1.14 (0.98–1.33)	0.093
3-month BNP (ng/L)	1.01 (1.00–1.01)	0.06		
3-month TMAO	1.28 (1.13–1.45)	<0.001	1.29 (1.13–1.50)	<0.001
Post-PCI LVEF (%)	1.07 (1.01–1.14)	0.015	1.07 (1.00–1.15)	0.057
Post-PCI LVEDV	1.00 (0.98–1.01)	0.605		
Post-PCI LAVI (mL/m^2^)	1.01 (0.96–1.07)	0.657		
Post-PCI E/A ratio	3.18 (1.44–6.99)	0.004	4.11 (1.61–10.98)	0.004
Post-PCI E/e’	1.08 (0.96–1.22)	0.19		
Post-PCI TRpV	0.69 (0.25–1.92)	0.482		
Pain-to-door time (minutes)	1.00 (1.00–1.00)	0.233		
Door-to-balloon (minutes)	1.01 (0.99–1.02)	0.599		
Culprit Vessel	0.47 (0.17–1.33)	0.154		
Number of diseased vessels	1.01 (0.50–2.03)	0.985		
Stent diameter (millimeters)	0.89 (0.41–1.94)	0.773		
Stent length (millimeters)	1.04 (0.98–1.11)	0.239		

PCI = Percutaneous Coronary Intervention, CRP = C-reactive protein, BNP = brain natriuretic peptide, LVEF = left ventricular ejection fraction.

## Data Availability

The datasets used in this study are available from the corresponding author on reasonable request.

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
