# Peer review of "Trimethylamine N-Oxide as a Biomarker for Left Ventricular Diastolic Dysfunction and Functional Remodeling After STEMI"

_ijms, 2025, doi:10.3390/ijms26073400_

Round 1

Reviewer 1 Report

Comments and Suggestions for Authors

The authors investigated TMAO as a possible marker for predicting FLVR following STEMI due to already known correlation between high TMAO levels and heart failure. They present data by following 204 patients. In the affected group TMAO levels increased significantly from baseline within the first 3 months.

In general, the manuscript is written-well and the data are in line with their initial hypothesis of using TMAO as a biomarker. I have two main comments:

  1. In the experimental part the authors should add information regarding the TMAO measurement in plasma. How it was carried out and any normalisation of the data.
  2. On page 10, under section 3.5 limitations, the authors mention the dietary effects. This is a major point since TMAO is produced in the gut by the conversion of choline containing foods such as meat, eggs and fish. In addition, the hepatic FMO3 is also responsible for the production of TMAO. The data presented and its significance can be changed if the baseline TMAO levels are altered due to diet and/or enzymatic conversions. These additional direct effects on TMAO should be emphasised in this section with the introduction of a few relevant references. 

Finally, on page 7 lines 83-85, I believe the authors have forgotten to delete these instructions which are present in the original template.

Author Response

  1. In the experimental part the authors should add information regarding the TMAO measurement in plasma. How it was carried out and any normalisation of the data.

Reply: We appreciate the comments from the reviewer. Please find the detailed description of the laboratory method below for the perusal of the reviewer.

Plasma samples were obtained from study participants through standard venipuncture techniques performed by trained medical staff. Whole blood was collected into EDTA-containing tubes to prevent coagulation. Within 30 minutes of collection, samples were centrifuged at 1,500 × g for 10 minutes to separate plasma from cellular components. The resulting plasma was carefully aliquoted into labeled microcentrifuge tubes to avoid hemolysis and contamination, then immediately stored at −80°C until further analysis to preserve the stability of target analytes.

Pasma levels of trimethylamine N-oxide (TMAO) were quantified using a commercially available enzyme-linked immunosorbent assay (ELISA) kit (Bioassay Technology Laboratory, Shanghai, China), following the manufacturer's instructions. 

Details of the calibration were added in to methods:

“For quantification, a standard calibration curve was generated using serial dilutions of known TMAO concentrations provided in the kit. Absorbance was measured at the recommended wavelength using the Infinite® M Plex microplate reader, and plasma TMAO concentrations were determined by interpolating the sample optical density values against the standard curve. All samples and standards were run in duplicate to ensure analytical precision, and results were expressed in µmol/L (µM)”

  1. On page 10, under section 3.5 limitations, the authors mention the dietary effects. This is a major point since TMAO is produced in the gut by the conversion of choline containing foods such as meat, eggs and fish. In addition, the hepatic FMO3 is also responsible for the production of TMAO. The data presented and its significance can be changed if the baseline TMAO levels are altered due to diet and/or enzymatic conversions. These additional direct effects on TMAO should be emphasised in this section with the introduction of a few relevant references. 

Reply: We appreciate the reviewer’s insightful observation. As noted in the limitations section of our discussion, we did not collect detailed dietary data, including changes in dietary patterns following the index STEMI. However, our study population is relatively homogeneous, comprising exclusively white Lithuanian individuals, which may reduce dietary variability.

We acknowledge that dietary intake significantly influences plasma TMAO levels. Consumption of foods rich in choline and carnitine, such as red meat, eggs, and fish, has been shown to elevate TMAO concentrations. For instance, studies have demonstrated that individuals consuming fish or red meat have a higher likelihood of elevated TMAO levels compared to those who abstain from these foods.[ Eur J Nutr. 2022 Feb 3;61(5):2357–2364.]  Additionally, the hepatic enzyme flavin-containing monooxygenase 3 (FMO3) plays a crucial role in converting dietary-derived trimethylamine into TMAO, further influencing systemic TMAO levels.

Given these factors, we recognize that the observed increase in TMAO levels from baseline to 3 months may be partially attributable to dietary influences. Future studies would benefit from incorporating comprehensive dietary assessments to better elucidate the relationship between dietary patterns and TMAO dynamics in this patient population.

In the revised manuscript, we added the following content to the limitation seciont:

Food rich in choline and carnitine, such as red meat, eggs, and fish, has been shown to elevate TMAO concentrations.[ Clin Kidney J. 2023 Apr 21;16(11):1804-1812., Eur J Nutr. 2022 Feb 3;61(5):2357–2364. ] Additionally, genetic variation in the hepatic enzyme flavin-containing monooxygenase 3 (FMO3) which plays a crucial role in converting dietary-derived trimethylamine into TMAO, may further influencing systemic TMAO levels.[ Sci Rep. 2019 Aug 12;9(1):11647.] However, our study population is relatively homogeneous, comprising exclusively white Lithuanian individuals, which may reduce dietary variability.

  1. Finally, on page 7 lines 83-85, I believe the authors have forgotten to delete these instructions which are present in the original template.

Reply: We thank the reviewer for spotting this proof reading mistake. This has been eliminated in the revised manuscript.

Reviewer 2 Report

Comments and Suggestions for Authors

Overall, this is an interesting paper. A few comments below for authors to consider. 

  1. In your study population - you focused just on STEMI patients with multivessel disease, which I think is fine, but does this limit how applicable your findings are to regular STEMI patients? Nearly 40% of your patients were female which seems higher than most STEMI studies - was this intentional or just how recruitment worked out?
  2. Interestingly, the baseline TMAO was not predictive compared with 3-month TMAO. You mention aspirin loading as a possible reason. Have other researchers seen similar patterns with TMAO after STEMI? If there are similar observations, what would be the reason behind?.
  3. The study validation cohort seems small (only 31 pts). How did your team decide on this number - is it enough for statistical power? And were these patients similar to your main group in terms of baseline characteristics?
  4. In Table 1, it is noticed that quite high rates of meds were administered at 3 months (95% on CCBs). Will this heavy treatment affect both TMAO levels and the development of LVDD/FLVR? Were there any changes in medication regime between 3-12 months that could have impacted your results?
  5. TMAO levels may be are hugely affected by diet - especially red meat, fish, etc. Did this current study collect any data on patients diets or try to control for this? This seems important given the big jump in TMAO from baseline to 3 months.
  6. The association between TMAO and remodeling is clear, but it not tell us much about the causality. Could not this high TMAO be a result of early cardiac dysfunction that was not caught in your baseline measurements?Please add some insights about the discussion of mechanistic studies.
  7. Finally, I am curious what is the actual clinical takeaway for this study? The paper suggest TMAO could identify high-risk patients, but what would we do differently for them? Especially since you mentioned the BIOSTAT-CHF study showed guideline therapies do not impact TMAO levels. What other approaches could we try for patients with high TMAO?

Author Response

Comments and Suggestions for Authors

Overall, this is an interesting paper. A few comments below for authors to consider. 

  1. In your study population - you focused just on STEMI patients with multivessel disease, which I think is fine, but does this limit how applicable your findings are to regular STEMI patients? Nearly 40% of your patients were female which seems higher than most STEMI studies - was this intentional or just how recruitment worked out?

Reply: We thank the reviewer for this thoughtful observation. Our focus on STEMI patients with multivessel disease was both scientifically and practically motivated. Clinically, this subgroup has a significantly higher risk of heart failure and less predictable recovery following primary PCI [ESC Heart Fail. 2020;8(1):222–237; Eur Heart J Suppl. 2024;26(Suppl 1):i39–i43]. Despite the established benefits of complete revascularization in these patients, their outcomes remain suboptimal, warranting further investigation into improved treatment strategies.

From a logistical perspective, in Lithuania, repeat catheterizations are not reimbursed unless clinically indicated, which made our study design more feasible if conducted in patients with multivessel disease.

Regarding the higher proportion of female patients (nearly 40%), this was not intentional but rather a result of consecutive patient enrollment. While unplanned, we are encouraged by the representation of women in our study, given their frequent underrepresentation in cardiovascular research.

  1. Interestingly, the baseline TMAO was not predictive compared with 3-month TMAO. You mention aspirin loading as a possible reason. Have other researchers seen similar patterns with TMAO after STEMI? If there are similar observations, what would be the reason behind?.

Reply: We appreciate the reviewer’s insightful observation. While TMAO has been infrequently reported in clinical studies involving STEMI patients, our literature review has identified studies that corroborate our findings.

For instance, Matsuzawa et al. [ Sci Rep. 2019 Aug 12;9(1):11647.] investigated the sequential change in TMAO levels in response to secondary prevention therapies in STEMI patients. They measured plasma TMAO levels at the onset of STEMI and 10 months later, finding that TMAO levels significantly increased from the acute phase to the chronic phase (median: 5.63 to 6.76 μM, P = 0.048). Importantly, the chronic-phase TMAO level independently predicted future cardiovascular events, whereas the acute-phase TMAO level did not. This suggests that post-STEMI TMAO levels may have greater prognostic value than baseline levels. Similarly, Almesned et al. [ J Clin Med. 2021 Dec 1;10(23):5677.] investigated TMAO levels at admission, 24 hours, and 4 months post-PCI in 379 STEMI patients. They observed that TMAO levels decreased from admission to 24 hours and then increased from 24 hours to 4 months.

These studies indicate that TMAO levels fluctuate during the post-STEMI period and that chronic-phase TMAO levels may be more predictive of cardiovascular outcomes than baseline levels.

Beyond the resumption of food intake and the impact of aspirin that we discussed in the manuscript, other potential explanations for this pattern include:

  1. Gut Microbiota Alterations: The gut microbiome, responsible for TMAO production, may change due to factors like stress, or dietary modifications during hospitalization and recovery.[ Nat Commun. 2023 Nov 9;14(1):7249.]
  2. Renal Function: As TMAO is renally excreted, fluctuations in kidney function post-STEMI, as a results of cardiorenal syndrome, develops later on in the disease course and can affect TMAO clearance.
  3. Medications: Beyond aspirin, the use of statin can also influence TMAO level at long-term follow-up. [Am J Cardiol. 2022 Sep 1:178:26-34.]

In the revised manuscript, we have modified the discussion as follows:

“We believe the causes are multiple. 1. In our cohort, baseline TMAO levels were low, aligning with the findings of Almesned et al. and Matsuzawa et al., likely due to fasting, vomiting, and dehydration during acute illness status. 2. Notably, all STEMI patients were given aspirin loading doses and statin which were known to reduce plasma TMAO levels. 3. The patient’s microbiota and metabolism may continue to evolve after the acute event as shown in previous studies in STEMI patients.”

  1. The study validation cohort seems small (only 31 pts). How did your team decide on this number - is it enough for statistical power? And were these patients similar to your main group in terms of baseline characteristics?

Reply: We thank the reviewer’s thoughtful statistical review. Our study is a sub study of the prospective, observational TAMIR study (NCT05406297) thus is not prospectively design to achieve a statistical powered sample size. That being said, to detect an anticipated Area Under the Receiver Operating Characteristic (ROC) Curve (AUC) of 0.7, with the null hypothesis being that the AUC equals 0.5, which corresponds to a test with no diagnostic ability beyond chance. Assuming a FLVR/LVDD prevalence of 20% in the population, a Type I error (alpha) of 0.025, and a desired power of 0.90 to correctly reject the null hypothesis when it is false, only 138 cases are needed for the study. Our study is thus having sufficient statistical power to prove our primary hypothesis. The acquisition of external validation is to establish the generalizability of the study not to further improve the statistical power. As described in the manuscript, the validation cohort were patients who met the same inclusion and exclusion criteria of the main study and were enrolled between August 2022 to February 2023. The baseline characteristics were unfortunately not fully collected for the validation cohort. However, their age and gender distribution are comparable with the main population as shown in the table below:

Group

N

Mean Age (SD)

Male (%)

Original

210

66.2 (12.3)

59.50%

Validation

31

67.2 (9.7)

64.50%

0.623

0.739

  1. In Table 1, it is noticed that quite high rates of meds were administered at 3 months (95% on CCBs). Will this heavy treatment affect both TMAO levels and the development of LVDD/FLVR? Were there any changes in medication regime between 3-12 months that could have impacted your results?

Reply:  We appreciate the reviewer’s careful analysis of our data. While it’s plausible that cardiovascular medications, including CCBs, might influence TMAO levels and cardiac remodeling, we observed that both the high vs. low TMAO groups as well as the LVDD/FLVR vs no LVDD/FLVR group had comparable medication regimens, as detailed in table 1 and the supplementary table below. Furthermore, all patients were consistently managed by the same clinical team, ensuring minimal changes in medication regimens between the 3-month and 12-month follow-up periods.

Baseline characteristic

TMAO quartile 1 (<0.51 µM)

N=53

TMAO quartile 2 (0.51~1.44 µM)

N=52

TMAO quartile 3 (1.44~3.36 µM)

N=51

TMAO quartile 4 (>3.36 µM)

N=48

p-value

Baseline medications

Beta-blockers, n (%)

8 (15.1)

7 (13.5)

7 (14.0)

10 (20.4)

0.767

Calcium channel blocker, n (%)

9 (17.0)

11 (21.2)

14 (28.0)

6 (12.2)

0.238

ACEI/ARB, n (%)

13 (24.5)

16 (30.8)

19 (38.0)

14 (29.2)

0.521

Aldosterone antagonists, n (%)

0 (0.0)

1 (1.9)

1 (2.0)

0 (0.0)

0.568

Statins, n (%)

9 (17.0)

9 (17.3)

8 (16.0)

9 (18.4)

0.992

Aspirin, n (%)

12 (23.1)

11 (21.2)

9 (18.0)

10 (20.4)

0.938

3-month follow-up medications

Beta-blockers, n (%)

48 (90.6)

43 (82.7)

43 (86.0)

42 (85.7)

0.705

Calcium channel blocker, n (%)

2 (3.8)

4 (7.7)

3 (6.0)

2 (4.1)

0.798

ACEI/ARB, n (%)

48 (90.6)

52 (100.0)

48 (96.0)

46 (93.9)

0.154

Aldosterone antagonists, n (%)

11 (20.8)

10 (19.2)

8 (16.0)

10 (20.4)

0.928

Statins, n (%)

51 (96.2)

50 (96.2)

48 (96.0)

48 (98.0)

0.944

Aspirin, n (%)

53 (100.0)

52 (100.0)

50 (100.0)

49 (100.0)

1

Ticagrelor, n (%)

41 (77.4)

47 (90.4)

46 (92.0)

43 (87.8)

0.116

Clopidogrel, n (%)

12 (22.6)

5 (9.6)

6 (12.0)

6 (12.2)

0.227

  1. TMAO levels may be are hugely affected by diet - especially red meat, fish, etc. Did this current study collect any data on patients diets or try to control for this? This seems important given the big jump in TMAO from baseline to 3 months.

Reply: We appreciate the reviewer’s insightful observation. As noted in the limitations section of our discussion, we did not collect detailed dietary data, including changes in dietary patterns following the index STEMI. However, our study population is relatively homogeneous, comprising exclusively white Lithuanian individuals, which may reduce dietary variability.

We acknowledge that dietary intake significantly influences plasma TMAO levels. Consumption of foods rich in choline and carnitine, such as red meat, eggs, and fish, has been shown to elevate TMAO concentrations. For instance, studies have demonstrated that individuals consuming fish or red meat have a higher likelihood of elevated TMAO levels compared to those who abstain from these foods.[ Eur J Nutr. 2022 Feb 3;61(5):2357–2364.]  Additionally, the hepatic enzyme flavin-containing monooxygenase 3 (FMO3) plays a crucial role in converting dietary-derived trimethylamine into TMAO, further influencing systemic TMAO levels.

Given these factors, we recognize that the observed increase in TMAO levels from baseline to 3 months may be partially attributable to dietary influences. Future studies would benefit from incorporating comprehensive dietary assessments to better elucidate the relationship between dietary patterns and TMAO dynamics in this patient population.

In the revised manuscript, we added the following content to the limitation seciont:

Food rich in choline and carnitine, such as red meat, eggs, and fish, has been shown to elevate TMAO concentrations.[ Clin Kidney J. 2023 Apr 21;16(11):1804-1812., Eur J Nutr. 2022 Feb 3;61(5):2357–2364. ] Additionally, genetic variation in the hepatic enzyme flavin-containing monooxygenase 3 (FMO3) which plays a crucial role in converting dietary-derived trimethylamine into TMAO, may further influencing systemic TMAO levels.[ Sci Rep. 2019 Aug 12;9(1):11647.] However, our study population is relatively homogeneous, comprising exclusively white Lithuanian individuals, which may reduce dietary variability.

  1. The association between TMAO and remodeling is clear, but it not tell us much about the causality. Could not this high TMAO be a result of early cardiac dysfunction that was not caught in your baseline measurements? Please add some insights about the discussion of mechanistic studies.

Reply: We appreciate the reviewer’s insightful observation regarding the relationship between TMAO levels and cardiac remodeling. While our study establishes a correlation between elevated TMAO levels and adverse cardiac remodeling, it does not establish causation. The possibility that increased TMAO concentrations result from early, undetected cardiac dysfunction is a valid consideration.

Several studies have demonstrated the direct impact of TMAO on cardiac myocyte function. A study demonstrated that elevated TMAO concentrations acutely increased contractile force in both human and mouse cardiac tissues. This effect was associated with increased intracellular calcium levels. However, the long-term exposure to increased intracellular calcium may induce excitotoxicity and cardiac damage.[ Am J Physiol Heart Circ Physiol. 2020 Apr 3;318(5):H1272–H1282.]

Several animal studies have shown that TMAO can induce cardiac hypertrophy and fibrosis through specific signaling pathways. For example, one study demonstrated that TMAO induces cardiac hypertrophy and fibrosis involving Smad3 signaling. These findings suggest a potential mechanistic link between elevated TMAO levels and adverse structural changes in the heart.[ Lab Invest. 2019 Mar;99(3):346-357.]

Conversely, it is logically feasible that it is early undetected cardiac dysfunction that leads to elevated TMAO levels.

  1. Finally, I am curious what is the actual clinical takeaway for this study? The paper suggest TMAO could identify high-risk patients, but what would we do differently for them? Especially since you mentioned the BIOSTAT-CHF study showed guideline therapies do not impact TMAO levels. What other approaches could we try for patients with high TMAO?

Reply: We deeply appreciate the reviewer for this insightful suggestion. We have added the following paragraph in the manuscript describing potential treatment options to reduce the impact of TMAO in CAD patients.

“Several therapeutic options have been developed to reduce circulating TMAO levels.

Dietary modifications: The mediterranean diet has been reported to reduce TMAO levels and the incidence of heart failure in large studies.[31,32] Similarly, a high-fiber diet or DASH diet could prevent the development of heart failure and improve cardiac remodeling.[33] Probiotics: certain strains have demonstrated cardioprotective effects by reducing LV hypertrophy, improving cardiac function, and decreasing inflammatory markers.[34] Prebiotics, such as inulin, can promote the growth of beneficial bacteria, improve insulin sensitivity, and reduce systemic inflammatory markers in obese patients.[35] Microbial TMA-lyase inhibitors: compounds such as fluoromethylcholine (FMC), iodomethylcholine (IMC), and 3,3-Dimethyl-1-butanol (DMB) reduce TMAO levels in animal models without affecting the viability of the symbiotic bacteria.[36,37] Natural phytochemicals such as allicin, resveratrol, indole-3-carbinol, berberine and 3,3ʹ-diindolylmethane could potentially reduce TMAO formation, but the potential adverse effects necessitate the need for careful clinical studies.[20] Fecal microbial transplantation (FMT) from lean donors has shown promise in improving the metabolic status of patients but  has never been tested in the heart failure population.[38]  Lastly, short-term antibiotic therapy, such as orally administered vancomycin, has been shown to reduce infarct size in animal models and was demonstrated to inhibit TMAO synthesis in humans.[39,40] However, this protective effect must be weighed against potential toxicity and the risk of drug-resistant microbiota. These strategies offer novel treatment avenues for heart failure, although further research is warranted to fully elucidate their mechanism and to optimize their clinical application.”